# Numerical Modeling of Submarine Pipeline Scouring under Tropical Storms

**Panyang Huang [1], Xin Meng [2], Haiyang Dong [3],*** and Lin Chong [3]

1 Second Institute of Oceanography, Ministry of Natural Resources, Hangzhou 310012, China; syhjjhpy@163.com
2 Ocean College, Zhejiang University, Hangzhou 310058, China; 21934107@zju.edu.cn
3 College of Architecture and Civil Engineering, Zhejiang University, Hangzhou 310058, China; 11812073@zju.edu.cn
* Correspondence: ocean0458@163.com or 11434023@zju.edu.cn; Tel.: +86-180-4230-1052

**Abstract:** Submarine pipelines are the lifelines of the national economy. Under the influence of typhoons, high-speed currents and waves continuously erode the seabed, leading to suspension or even rupture of pipelines. Therefore, it is of great importance to study the sediment transport under the action of waves and currents. A numerical model of sediment scouring and deposition combining wave and currents is established, which considered tidal current, storm surges, wind waves, and sediments in the East China Sea. Combining with the monitoring of the actual laying condition of pipelines, it is found that the area with the most serious scouring is around KP300. It is shown that the typhoon weather with high intensity and density will lead to the suspension of pipelines, which is noteworthy in the construction of marine engineering.

**Keywords:** sediment transport; numerical model; typhoon; pipelines; local scour





## 1. Introduction

Oil and gas pipelines are regarded as the lifelines of marine energy transmission. Since the first submarine pipeline was laid in the Mexico Gulf by the United States in 1954, the bottom of the sea has seen vigorous development of submarine pipelines all over the world [1]. Nowadays, more than 3000 km of submarine pipelines have been laid in China. Along the coast of Zhejiang Province, submarine pipelines of groups of Chunxiao Gas Field, Pinghu Oil and Gas Field and Hangzhou Bay Cross-sea Oil have been constructed [2].

The pipeline of Pinghu Oil and Gas Field landed in Zhoushan City, Zhejiang Province where two pipelines broke on 2000 (Chen et al. 2005) [3], leading to oil spills. As a result, hundreds of millions of dollars were spent, and serious environmental damage was caused by widespread oil pollution. Therefore, it is essential to ensure the safety of submarine pipelines. Figure 1 shows how accidents happened due to the local scour at pipelines.

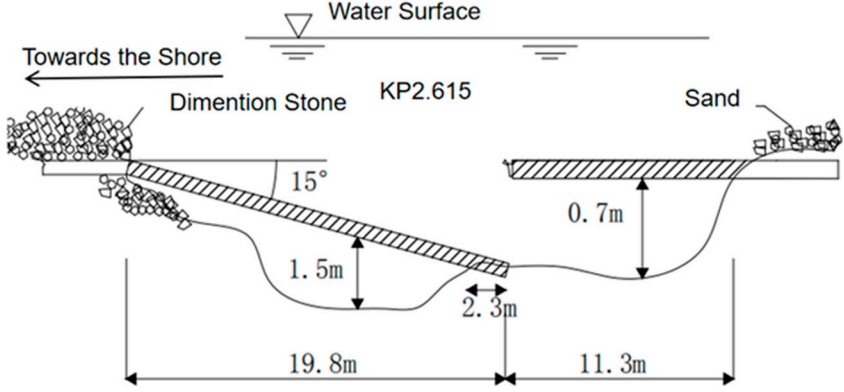

**Figure 1.** Pinghu submarine oil and gas pipeline broke in 2000.

In 1982, six European leading gas storage and transport companies launched a campaign to collect data on accidents in pipeline systems, setting up the European Pipeline Accident Data Organization (EGIG), which generated statistics about the causes of the failure of submarine gas pipelines between 1970 and 2010 [4]. Now, EGIG is a co-operation between a group of seventeen major gas transmission system operators in Europe and is the owner of an extensive gas pipeline-incident database. The creation of this extensive pipeline-incident database (1982) [5] has helped pipeline operators to demonstrate the safety performances of Europe's gas pipelines, which is useful for the pipeline operators to improve safety in their gas pipeline transmission systems [6] (EGIG,2018).

Scouring at pipelines is affected by many environmental factors such as topography, submarine sediments, marine hydrology and wind fields [7–11]. The strong hydrodynamic disturbance, which is caused by typhoons, destroys the equilibrium state of seabed sediments under the action of a tidal wave, which usually takes a long time to form. At the peak of the strongest storm action, the currents strongly scour the seabed, carrying a large amount of sediment in water, which is transported to the sea with the ebb tide of the storm surge, thus affecting the laying state of the undersea pipelines [2,5,12,13].

The application of numerical simulations in scouring and silting, such as the shear stress calculation model of horizontal seabed surface model of sediment transport under storm conditions, has increased tremendously in recent years for analyzing different environmental factors [14–17]. These mathematical models simulate the scouring and silting situation near the pipeline under certain natural factors, which can reveal the local scouring and silting law. However, there are a few studies on the sediment module based on the wave–current coupling model, and the changes of waves, tidal currents and sediment caused by typhoons have not been fully discovered.

Western European countries began to study the third generation of wave numerical models in 1985, and a WAM group was established to develop a new wave numerical model with a more comprehensive consideration of the source functions in the energy balance equation. Among them, the WAM model and WAVEWATCH III model are the representatives of the third generation of wave numerical models [18]. Later, Booij summarized and modified the third-generation model, especially the results of WAM parameterization, and proposed SWAN, which is suitable for the computation of offshore waves. After continuous improvement, SWAN has been widely applied in the field of offshore marine science and engineering. Although the same wave model is the third generation, there are also differences in performance and applicability. In order to foster strengths and circumvent weaknesses, nested applications of different modes have appeared [19]. For example, using WAVEWATCH III to build the global wind and wave model and nested SWAN model in the local nearshore area, SWAN can improve the accuracy after obtaining better boundary data. In the simulation of near-shore wind and wave, SWAN often has a better effect than WAVEWATCH III.

In this research, a large area numerical model was established with the consideration of the interaction between wave and water flow, which simulated the dynamic elements of the ocean under a typhoon. Meanwhile, the scouring and silting conditions of the offshore area with complicated topography during a typhoon was considered, and the rules of scouring and silting in the area where the submarine pipeline is studied. Considering currents and waves, the numerical model is used to study the safety of submarine pipelines in specific areas and specific extreme natural conditions. This new method can provide a sound basis for line selection of submarine pipelines under extreme conditions, such as complex terrain, geomorphologic region and hydrodynamic environment, and can also provide reference for the monitoring of submarine pipelines.

## 2. Materials and Methods

### 2.1. Study Area

As is shown in Figure 2, the Chunxiao Gas Field Group (28°10′ N~28°48′ N, 124°45′ E ~125°15′ E) is located in the East China Sea continental shelf, which is about 450 km to

the southeast of Shanghai city and 350 km to the south of Ningbo city, Zhejiang province. In 2007, after Typhoon Lily, pipeline suspension was detected near KP300, thus section KP287.665~KP301.906 was selected as the digital-analog area. Several exposed and suspended sections were found in the offshore section of Chunxiao gas field group subsea pipeline, which seriously threatens the safety of the pipeline.

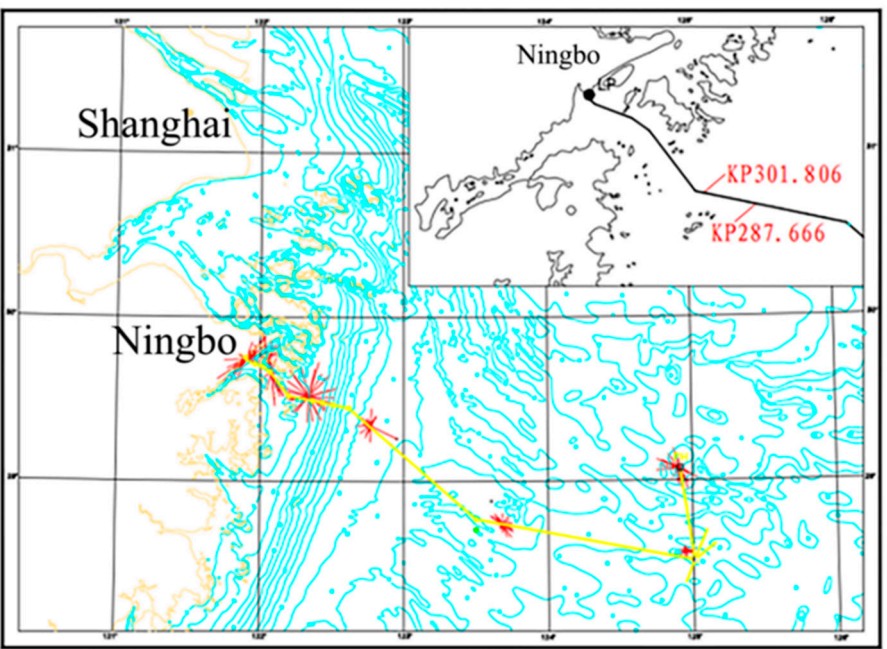

**Figure 2.** Location of Chunxiao Submarine Pipeline.

*2.2. Environmental Factors*

2.2.1. Topography

The entire seafloor where the submarine pipeline is located gradually inclines in the SE direction with a maximum water depth of 110 m and an average depth of 90~110 m. The topography of the seabed in the whole area is basically floating without severe undulations or steep ridges, whose average slope is $0.3 \times 10^{-3}$. There are tidal channels, underwater shore slopes, comb-shaped sand ridges and the central submarine plain areas of the continental shelf from the land to the sea.

The local topography has a great impact on the hydrodynamic environment. In this research, the KP287.66~KP301.906 section of the submarine pipeline is located on the outer edge of the Mountain Niubi water channel, which belongs to the underwater bank slope section where water depth is between 11 and 15 m. Besides, the sedimentary landform is the main feature in this area, and the seabed sediments are mainly deposited by the coastal rivers since the Holocene.

In this paper, the 1:250000 coastline data from National Geophysical Data Center of National Oceanic and Atmospheric Administration (NOAA) is selected (https://www.ngdc.noaa.gov/mgg/shorelines/shorelines.html accessed on 17 May 2021), whose resolution and precision can meet the requirements.

As is shown in Figure 3, the high-resolution water depth data accumulated by the Second Institute of Oceanography of the State Oceanic Administration for many years were used in the area with complex coastal topography, and 30″ × 30″ bathymetric data provided by the UK Ocean Data Center were used in offshore areas. (https://www.bodc.ac.uk/data/hosted_data_systems/gebco_gridded_bathymetry_data/ accessed on 17 May 2021).

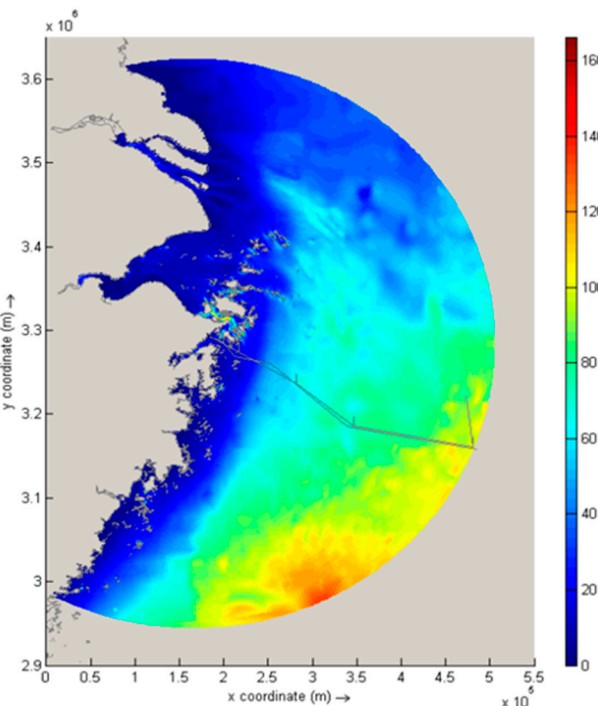

**Figure 3.** Sea topography and coastline.

### 2.2.2. Seafloor Sediments

The subsoil in the study area can be roughly divided into clayey soil, silty soil and sandy soil, whose spatial distribution is shown in Figure 4. The seabed is mainly clay soil from KP287.66 to KP301.906. The offshore sea is sandy soil, and the silts mainly appear as interlayers in the sea area near KP150. In 2007, the sea area with pipeline suspension was monitored. The sediment was mainly composed of silt and clay, of which clay was slightly dominant, accompanied by a small amount of shell fragments, and the sediment particle size was between 5 and 75 μm.

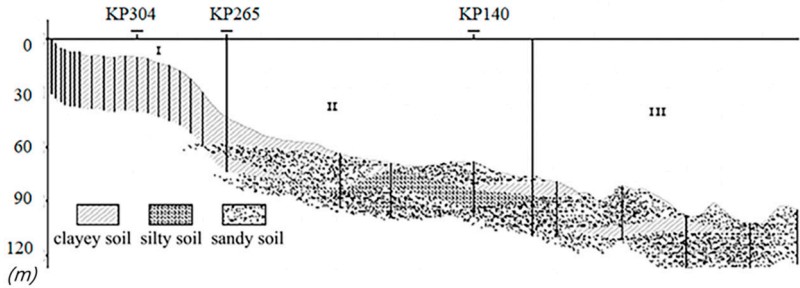

**Figure 4.** Zonation map of submarine pipeline routing area.

### 2.2.3. Marine Hydrology

According to measured data, the tidal current in the pipeline area is typical irregular semi-diurnal. The measured maximum velocity of the bottom layer was 96 cm/s and the residual velocity was 2.2 cm/s. Moreover, the flow rate of high tide is slightly higher than that in low tide. The movement of the tidal current is typical reciprocating flow near the shore, showing a rotating flow in the outer sea area (outside the Mountain Niubi water channel). Generally, the coastal waters of Zhejiang are mainly wind-generated waves which are more sensitive to typhoons and not controlled by the incoming surges outside the wind-zone.

2.2.4. Wind Field

Statistics in recent decades show that the average number of tropical cyclones affecting the region is about four times per year. The years that were most affected by tropical cyclones were 1959 and 1960, respectively, while the year that was least affected was 1996. Tropical cyclones generally occur from May to November, which are mainly concentrated from July to September, accounting for about 80% of the total. August is the peak of tropical cyclone activity that accounts for 35% of the total.

*2.3. Local Scour Modeling*

2.3.1. Storm Surge Model

Under the assumption of incompressible liquid and shallow water, the Navier Stokes Equations were solved ignoring vertical acceleration. Orthogonal curvilinear grid was adopted for a horizontal plane (Arakawa C-type grid system is used for equation variable arrangement), while σ grid was applied for vertical direction which can fit the terrain. K-ε turbulence model was selected, and finite difference method was used to discrete the equation.

The vertical average mass conservation equation is as follows.

$$\frac{\partial \zeta}{\partial t} + \frac{1}{\sqrt{G_{\xi\xi}}\sqrt{G_{\eta\eta}}} \frac{\partial \left[(d+\zeta)U\sqrt{G_{\eta\eta}}\right]}{\partial \xi} + \frac{1}{\sqrt{G_{\xi\xi}}\sqrt{G_{\eta\eta}}} \frac{\partial \left[(d+\zeta)V\sqrt{G_{\xi\xi}}\right]}{\partial \eta} = Q \quad (1)$$

Q represents the role of source and sink, such as drainage and precipitation.
Momentum conservation equation in a direction $\zeta$:

$$\frac{\partial u}{\partial t} + \frac{u}{\sqrt{G_{\xi\xi}}}\frac{\partial u}{\partial \xi} + \frac{v}{\sqrt{G_{\eta\eta}}}\frac{\partial u}{\partial \eta} + \frac{uv}{\sqrt{G_{\xi\xi}}\sqrt{G_{\eta\eta}}}\frac{\partial \sqrt{G_{\xi\xi}}}{\partial \eta} - \frac{v^2}{\sqrt{G_{\xi\xi}}\sqrt{G_{\eta\eta}}}\frac{\partial \sqrt{G_{\eta\eta}}}{\partial \xi} - fv$$
$$= -\frac{1}{\rho_0 \sqrt{G_{\xi\xi}}}P_\xi + F_\xi + M_\xi + \frac{1}{(d+\zeta)^2}\frac{\partial}{\partial \sigma}\left(v_V \frac{\partial u}{\partial \sigma}\right) \quad (2)$$

Momentum conservation equation in a direction $\eta$:

$$\frac{\partial v}{\partial t} + \frac{u}{\sqrt{G_{\xi\xi}}}\frac{\partial v}{\partial \xi} + \frac{v}{\sqrt{G_{\eta\eta}}}\frac{\partial v}{\partial \eta} + \frac{uv}{\sqrt{G_{\xi\xi}}\sqrt{G_{\eta\eta}}}\frac{\partial \sqrt{G_{\eta\eta}}}{\partial \xi} - \frac{u^2}{\sqrt{G_{\xi\xi}}\sqrt{G_{\eta\eta}}}\frac{\partial \sqrt{G_{\xi\xi}}}{\partial \eta} + fu$$
$$= -\frac{1}{\rho_0 \sqrt{G_{\eta\eta}}}P_\eta + F_\eta + M_\eta + \frac{1}{(d+\zeta)^2}\frac{\partial}{\partial \sigma}\left(v_V \frac{\partial v}{\partial \sigma}\right) \quad (3)$$

In $\sigma$ coordinate, the vertical velocity component is solved by mass conservation equation:

$$\frac{\partial \zeta}{\partial t} + \frac{1}{\sqrt{G_{\xi\xi}G_{\eta\eta}}}\frac{\partial [(d+\zeta)]u\sqrt{G_{\eta\eta}}}{\partial \xi} + \frac{1}{\sqrt{G_{\xi\xi}G_{\eta\eta}}}\frac{\partial [(d+\zeta)]v\sqrt{G_{\xi\xi}}}{\partial \eta} + \frac{\partial \omega}{\partial \sigma}$$
$$= H(q_{in} - q_{out}) \quad (4)$$

Here, $\omega$ refers to the vertical velocity perpendicular to the $\sigma$ coordinate plane, which changes as the $\sigma$ coordinate plane moves up and down. In the above formulas, H is the water depth, $H = h + \zeta$ is the water level, and  is the water depth relative to the mean sea level; $G_{\xi\xi}$ and $G_{\eta\eta}$ are the conversion coefficients of curvilinear coordinate system to rectangular coordinate system; U and V are the average velocity in $\xi$ and $\eta$ direction respectively. $g$ is the acceleration of gravity; $f$ is the parameter of Coriolis force; $F_\xi$ and $F_\eta$ are turbulent momentum fluxes in $\xi$ and $\eta$ directions, respectively. $P_\xi$ and $P_\eta$ represent the water pressure gradient in $\xi$ and $\eta$ directions; $M_\xi$ and $M_\eta$ represent the source or sink of momentum in the $\xi$ and $\eta$ directions, respectively; $q_{in}$ and $q_{out}$ represent source and sink items.

ADI scheme is a common method for solving parabolic and elliptic equations in a finite difference method. It is improved from crank Nicolson scheme. It has second-order accuracy in time and space and is unconditionally stable. It divides a time step into

two steps and transforms a five diagonal linear equation system into a tridiagonal linear equation system, which greatly improves the computational efficiency and is widely used. In the horizontal direction, the model is discretized by the ADI method.

Step 1:

$$\frac{\overrightarrow{U}^{l+\frac{1}{2}} - \overrightarrow{U}^{l}}{\frac{1}{2}\Delta} + \frac{1}{2}A_x\overrightarrow{U}^{l+\frac{1}{2}} + \frac{1}{2}A_y\overrightarrow{U}^{l} + B\overrightarrow{U}^{l+\frac{1}{2}} = \overrightarrow{d} \tag{5}$$

Step 2:

$$\frac{\overrightarrow{U}^{l+1} - \overrightarrow{U}^{l+\frac{1}{2}}}{\frac{1}{2}\Delta} + \frac{1}{2}A_x\overrightarrow{U}^{l+\frac{1}{2}} + \frac{1}{2}A_y\overrightarrow{U}^{l+1} + B\overrightarrow{U}^{l+1} = \overrightarrow{d} \tag{6}$$

Among them:

$$A_x = \begin{pmatrix} 0 & -f & g\frac{\partial}{\partial x} \\ 0 & u\frac{\partial}{\partial x} + v\frac{\partial}{\partial y} & 0 \\ H\frac{\partial}{\partial x} & 0 & u\frac{\partial}{\partial x} \end{pmatrix} \tag{7}$$

$$A_y = \begin{pmatrix} u\frac{\partial}{\partial x} + v\frac{\partial}{\partial y} & 0 & 0 \\ f & 0 & g\frac{\partial}{\partial y} \\ 0 & H\frac{\partial}{\partial y} & v\frac{\partial}{\partial y} \end{pmatrix} \tag{8}$$

$$B = \begin{pmatrix} \lambda & 0 & 0 \\ 0 & \lambda & 0 \\ 0 & 0 & \lambda \end{pmatrix} \tag{9}$$

where $\lambda$ is the linear bottom friction coefficient, $\overrightarrow{d}$ is the external force, such as wind and atmospheric pressure. In the first step, the calculation process is from $l$ to $l + \frac{1}{2}$, that is, the calculating time is from $t = l\Delta t$ to $t = \left(l + \frac{1}{2}\right)\Delta t$. In this process, V Equation (3), U Equation (2) and continuity Equation (1) coupled by free surface gradient are solved; In the second step, the calculation process is from $l + \frac{1}{2}$ to $l + 1$, that is, the calculation time is from $\left(l + \frac{1}{2}\right)\Delta t$ to $(l + 1)\Delta t$. In this process, U Equation (2), V Equation (3) and continuity Equation (1) coupled by free surface gradient are solved.

### 2.3.2. Storm Wave Model

In SWAN, the wave spectrum is described by the spectral action balance equation given in Hesselmann [20]. The SWAN model does not use two-dimensional spectral density, but rather uses the two-dimensional dynamic spectral density to represent random waves. In the flow field, the dynamic spectral density is conserved, but the energy spectral density is not. The dynamic spectral density $N(\sigma, \theta)$ is the ratio of the energy spectral density $E(\sigma, \theta)$ to the relative frequency $\sigma$. In a Cartesian coordinate system, the dynamic spectrum balance equation is expressed as follows:

$$\frac{\partial}{\partial t}N + \frac{\partial}{\partial x}c_xN + \frac{\partial}{\partial y}c_yN + \frac{\partial}{\partial \sigma}c_\sigma N + \frac{\partial}{\partial \theta}c_\theta N = \frac{S}{\sigma} \tag{10}$$

In the formula, the first term on the left $\frac{\partial}{\partial t}N$ is the change of wave action with time, the second term is $\frac{\partial}{\partial x}c_xN$, the third term $\frac{\partial}{\partial y}c_yN$ is the propagation of wave action in geographical space, the fourth term $\frac{\partial}{\partial \sigma}c_\sigma N$ is Doppler shifting caused by topography and current, and the fifth term $\frac{\partial}{\partial \theta}c_\theta N$ is refraction caused by topography and current, Is the source term, including wind-induced wave, dissipation, nonlinear wave action and wave breaking. They are the propagation velocity in the direction. $S(= S(\sigma, \theta))$ is the source term, including wind-induced wave, dissipation, nonlinear wave action and wave breaking. $c_x, c_y, c_\sigma, c_\theta$ are propagation velocities in $(x, y, \sigma, \theta)$ direction. Various physical processes

considered in the model are input through the right source term, including wind energy input, energy dissipation and nonlinear wave interaction, the specific forms are as follows.

Wind energy input:

The input of wind energy is described by resonance mechanism feedback mechanism in SWAN model:

$$S_{in}(\sigma,\theta) = A + BE(\sigma,\theta) \tag{11}$$

The coefficients are determined by wave frequency and direction as well as wind speed and direction.

Energy dissipation:

The dissipation of wave energy is mainly composed of three parts, including white wave dissipation $S_{ds,w}(\sigma,\theta)$, bottom friction dissipation $S_{ds,b}(\sigma,\theta)$ and wave breaking dissipation $S_{ds,br}(\sigma,\theta)$.

The white wave is mainly controlled by the wave steepness. In the current third generation wave model, the white wave dissipation adopts the Hsselmann formula [20].

$$S_{ds,w}(\sigma,\theta) = -\Gamma\widetilde{\sigma}\frac{k}{\widetilde{k}}E(\sigma,\theta) \tag{12}$$

where $\Gamma$ is a coefficient related to wave steepness, $\widetilde{\sigma}$ and $\widetilde{k}$ are average wave frequency and average wave number.

The dissipation caused by topography is mainly caused by bottom friction, seafloor infiltration and reflection scattering of irregular topography. In the continental shelf of a sandy seabed, seabed dissipation is mainly caused by bottom friction, which can be expressed as follows:

$$S_{ds,b}(\sigma,\theta) = -C_{bottom}\frac{\sigma^2}{g^2 sin^2(kd)}E(\sigma,\theta) \tag{13}$$

where $C_{bottom}$ is the bottom friction coefficient.

For the dissipation caused by wave breaking, swan adopts the following formula:

$$S_{ds,br}(\sigma,\theta) = \frac{D_{tot}}{E_{tot}}E(\sigma,\theta) \tag{14}$$

For the dissipation caused by wave breaking, the following formula is used in SWAN: $E_{tot}$ is the total energy and $D_{tot} < 0$ is the total energy dissipation rate caused by wave breaking. It mainly depends on the wave breaking factor $\gamma = H_{max}$, which can be a constant or a variable in SWAN.

Nonlinear wave interaction:

Nonlinear wave interaction plays a very important role in wave deformation. In deep water, the fourth-order wave interaction determines the evolution of spectral frequency, which can transform wave energy from peak frequency to low-frequency and high-frequency. In shallow water, third-order wave interaction can transform wave energy from low-frequency to high-frequency to generate high-order harmonics. In the SWAN model, the discrete interaction approximation (DIA) and the centralized third-order approximation (LTA) are used to approximate.

Figure 5 shows the disturbed area of numerical calculation results in the model calculation domain under wind and wave conditions. The rectangle represents the calculation water area, the thick solid line represents the wave front boundary with known boundary conditions, the thin solid line represents the lateral boundary of the unknown wave, and the shaded area represents the disturbed area of the model result due to the assumption. In the case of wind wave, the disturbed region propagates from the top of the known wave boundary conditions to the shore in the range of 30–45° with the average wave direction. For the swell wave, the disturbed area will be smaller due to its smaller directional distribution range.

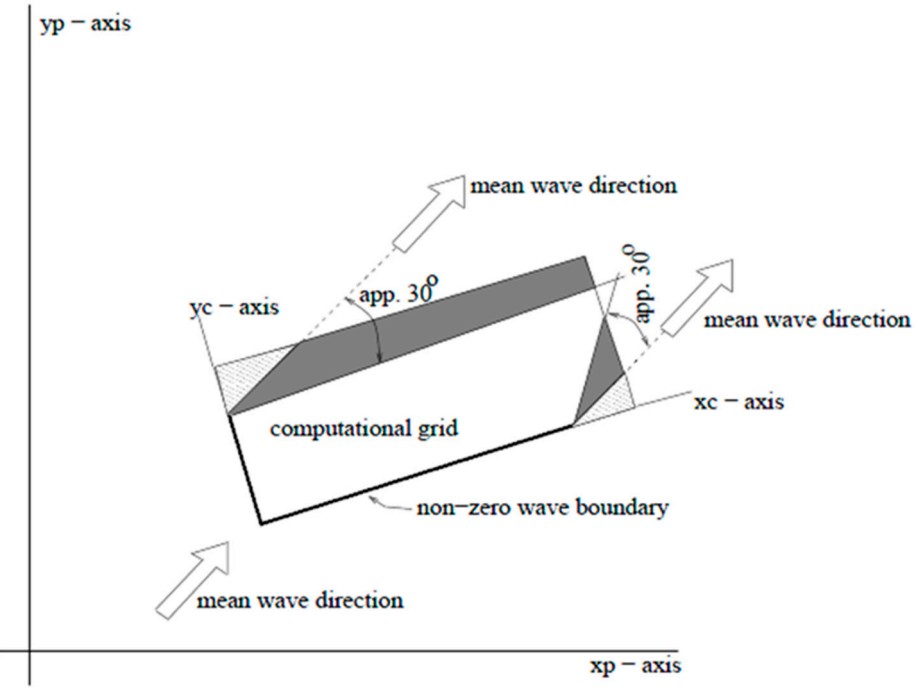

**Figure 5.** Schematic diagram of calculation distortion area caused by unknown boundary.

A typical Typhoon Lily in 2007 was selected to calculate the typhoon wave. Its path is shown in Figure 6. The same water depth and shoreline data are used as that in calculation of storm surge, and the calculation method of wind field is the same.

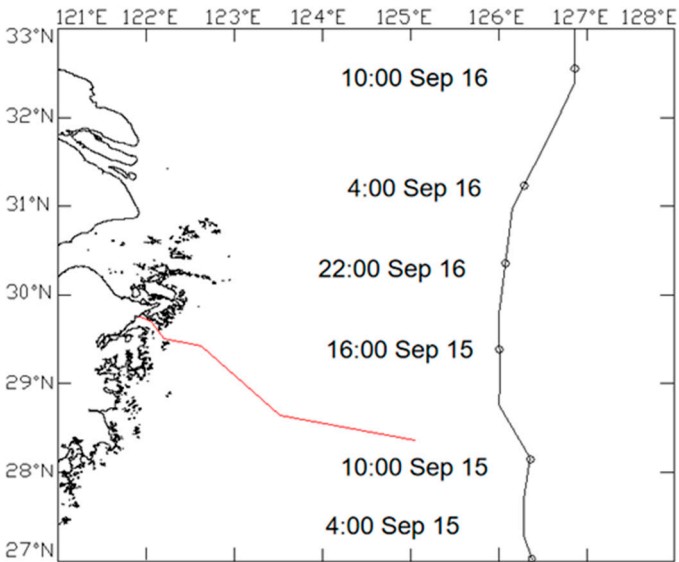

**Figure 6.** The path of Typhoon Lily (Beijing time).

The verification of calculation results is a difficult problem in wave numerical simulation. Generally, there are three methods: the first is to verify with the measured data, the second is the data collected by the satellite altimeter, and the third is the data provided by other recognized and reliable numerical wave prediction models. Obviously, it is the most ideal if it can be verified by the measured data, but generally there is little chance to obtain the wave data of fixed area and fixed time period. For satellite observation, the reliability is greatly lower than the measured data, and it may not be able to capture the data we just need for the satellite may not be running in the wave calculation time. For the data

provided by other models, the biggest drawback is the reliability of the data, especially the nearshore terrain.

Due to the limitation of conditions, the data provided by the global wave prediction model WAVEWATCH III of NOAA is used for verification. The resolution of the model is $1° \times 1.25°$ and there are special calculations for typhoons in some regions such as the Gulf of Mexico. The resolution of the model is relatively high because it is a targeted calculation, and its accuracy is relatively high. Unfortunately, in the East China Sea, this kind of study is not provided.

Figure 7 shows the distribution of global ocean effective wave height predicted by WAVEWATCH III at 14:00 Beijing time on 15 July 2007. Near the East China Sea (the area in the red box), the annular wave field caused by typhoon Lily is clearly discernible. Effective wave height refers to the actual wave height calculated according to certain rules. Since sea waves are actually a random combination of waves with different wave heights, periods and directions, the wave height value of a wave is not representative. For this reason, in any wave group composed of *n* waves, the wave heights in the wave train are arranged in order from large to small, and the first 1/3 waves are determined as effective waves. The wave height and period of the effective wave are equal to the average wave height and average period of N over three waves [19].

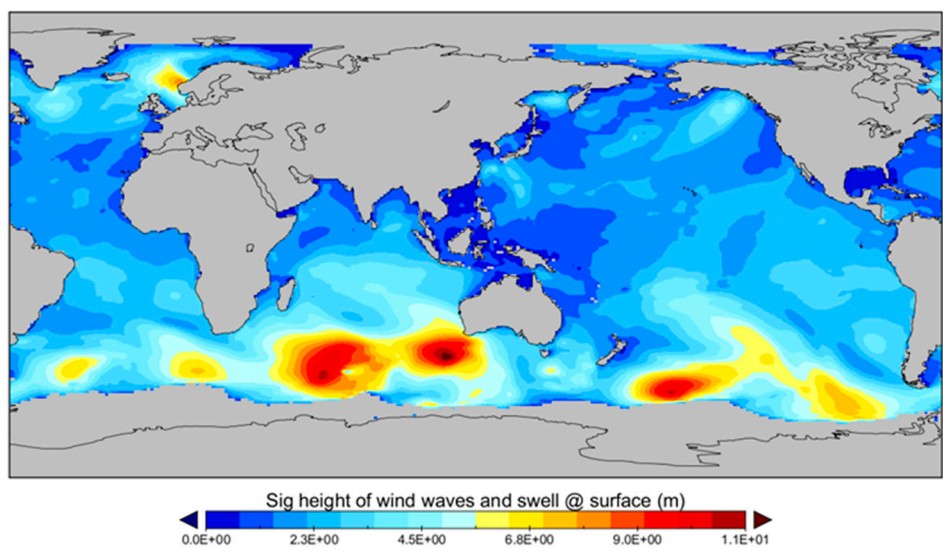

**Figure 7.** Effective wave height from WAVEWATCH III (14:00 15 July 2007 Beijing time).

In the calculation area of this paper, three effective points are selected for verification, and the verification results are shown in Figure 8. Although the two values are not in good agreement, the trend of significant wave height change is very consistent, and the peak values of three points appear at the same time. It shows that the model can reflect the response of waves in Zhejiang coastal area when typhoon passes by. The difference of the significant wave height between the two is caused by many reasons, some of which are even unavoidable. First of all, although they belong to the third generation of numerical wave models, there are differences in theory, especially in the nearshore, the processing of various physical processes is not the same, in addition, the input of wind energy is not consistent; secondly, the three boundaries of SWAN model in this paper cannot provide real boundary conditions, which will certainly affect the calculation accuracy; finally, the WAVEWATCH of NOAA in terms of the accuracy of typhoon analysis, is lower than that of the global wind field.

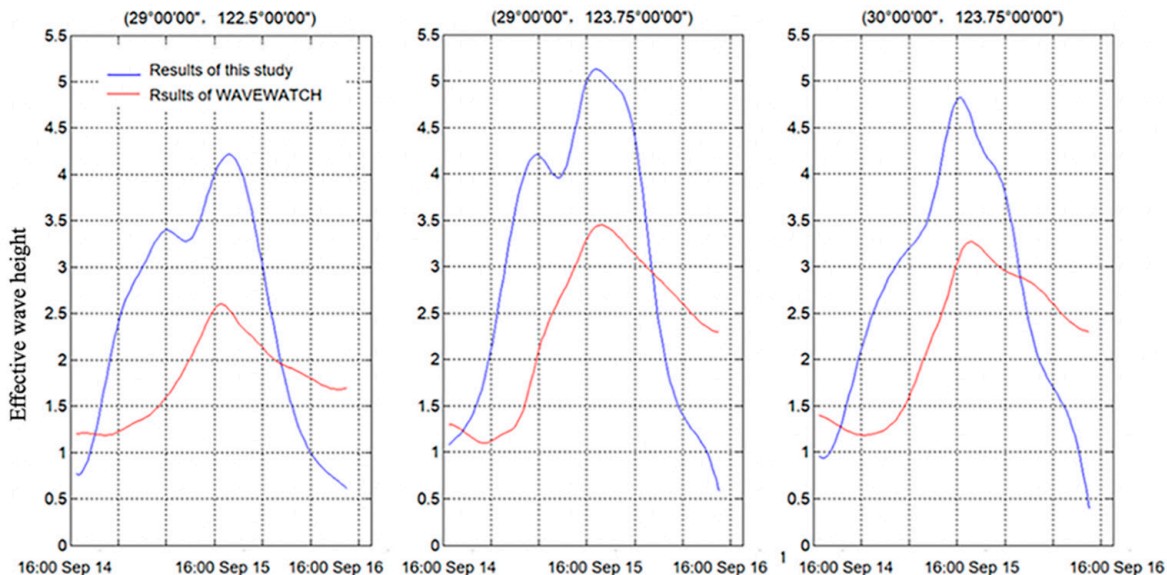

**Figure 8.** Verification results of effective wave height.

Comparing the significant wave heights at different locations, it is obvious that the points in the open sea (with greater longitude) have larger peaks, and the points in the South have larger peaks than those in the north. It can be seen from Figure 8 that the two calculation results are consistent.

Although the two lines of effective height do not correspond well, they are very consistent in terms of the trend of effective wave heights, and the peaks of the three points all appear at the same time. For this research, the ultimate purpose is not to study the platform wind and wave itself, but to consider the influence of waves on the seabed shear stress and scour effect. Therefore, such a result is acceptable. This shows that the model can reflect the wave response in the coastal area of Zhejiang when a typhoon passes by. In addition, the difference of effective wave heights between the two is also caused by many reasons, some of which are even unavoidable. First of all, although both belong to the third-generation numerical wave model, there are theoretical differences, especially in the nearshore, the processing of various physical processes is not the same, and the input of wind energy is not consistent. Secondly, none of the three boundaries of the SWAN model in this paper can provide real boundary conditions, which will certainly affect the calculation accuracy. Finally, NOAA's WAVEWATCH III forecast value is for the whole world, with a relatively low resolution. Due to the topography and other factors, the accuracy of the near shore is relatively poor. For the regions affected by the typhoon field, there is no detailed wind field, which is bound to affect the calculation accuracy.

Figure 9 shows the distribution of significant wave height at four typical moments in the calculation area under Typhoon Lily. The four stages are: (a) when the center is far away from the calculation area, the wind field has little influence on the region, so the wave is mainly controlled by the open sea boundary; (b) when the typhoon is close to the calculation area, the regional wave is obviously affected by the wind field, thus the effective wave height of the region affected by strong wind began to increase significantly; (c) after the effect of the wind field on the region has lasted for appropriately 10 h, the wave is significantly increased, and the typhoon center is closest to the center of the region, which is also the time when the typhoon field has the strongest impact on the entire region. At this time, the wave of the whole region reaches its peak; (d) the typhoon leaves slowly to the north, the wind field begins to weaken, and the wave also weakens.

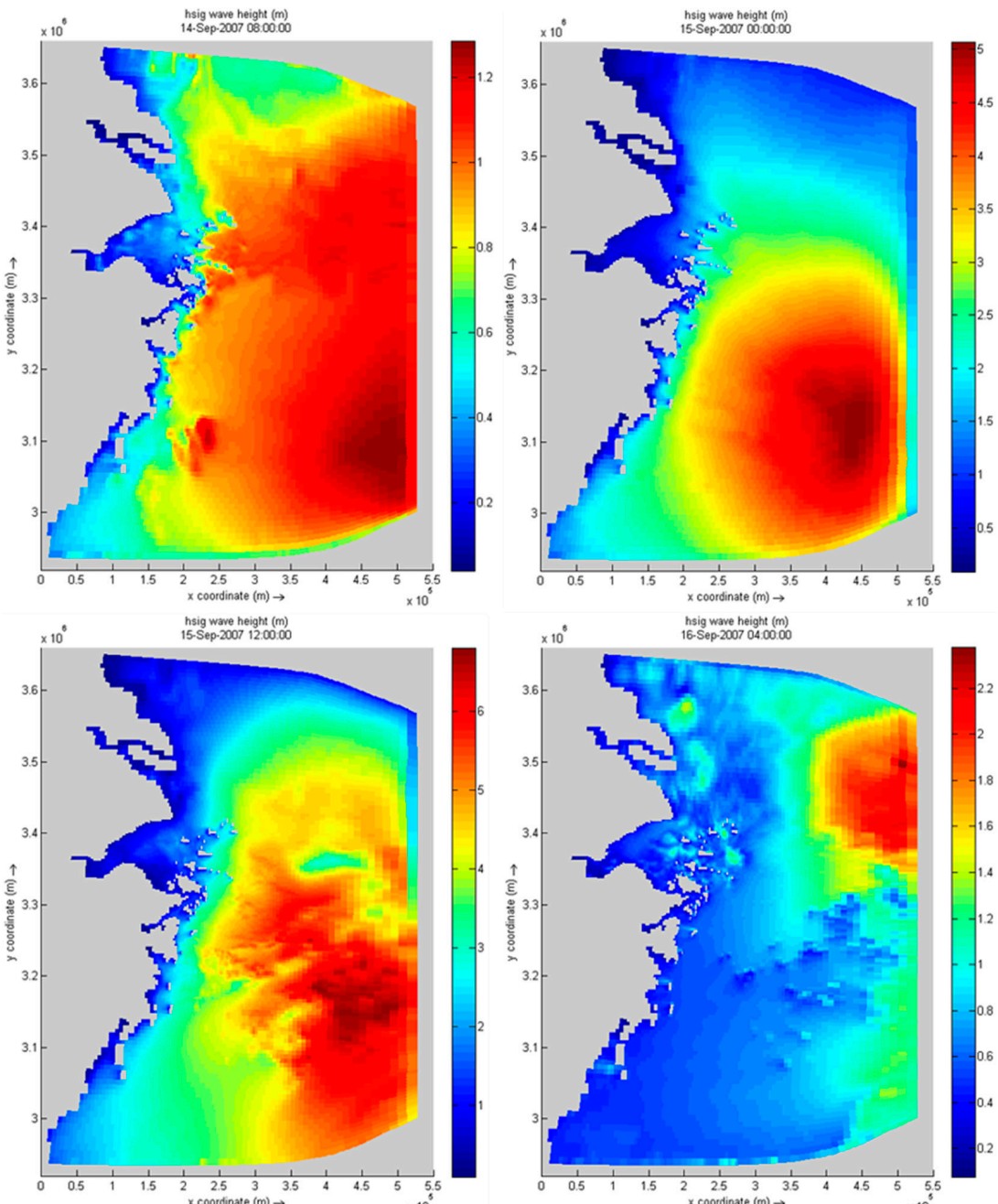

**Figure 9.** Distribution of effective wave height at four moments under Typhoon Lily.

It can also be seen from Figure 9 that although the nearshore is not directly affected by a strong typhoon field, the effective wave height is about 3 m due to the wave conduction from the open sea. In Zhoushan Archipelago, due to the blocking effect of many islands on waves, the wave height in the Western sea area of the islands is obviously smaller.

### 2.3.3. Coupling Model

At present, the wave elements are considered in the most commonly used numerical model for calculating the interaction of two-dimensional waves and currents based on the N-S equation to deduce the mathematical model. This model is similar to the two-dimensional tidal current control equations, with radiation stress terms, where the wave current interaction is included.

However, due to the computational efficiency and complexity of the model, most of the models only consider the effect of unidirectional wave elements on the flow field

and thus cannot consider the influence of the flow field on the wave field synchronously. Therefore, the wave current interaction is uncoupled or pseudo coupled, resulting in relatively lower simulation accuracy when the wave influence is large.

Driven by the boundary dynamic force, at first, the flow field calculation is carried out, outputting the elements affecting the wave to the wave calculation model, and then the wave field calculation of the first step is started. After that, the elements affecting the flow field are delivered to the flow field model for the calculation of the next step. This cycle is repeated until the calculation reaches the specified time.

2.3.4. Scouring Model

A coupled large-area model is established with consideration of ocean dynamic elements under typhoon, such as the interaction of waves and currents. Meanwhile, the scouring and silting of the offshore areas with complicated topography during typhoon was investigated, which help understand the scouring and silting rules of the area where submarine pipelines are located.

The sediment transport module was used to simulate the transport of cohesive and non-cohesive sediment (such as the diffusion of dredged material) and to study the sediment deposition/erosion model.

The appropriate boundary conditions are used to solve the two-dimensional convection-diffusion equation. In sediment calculation, the two-dimensional diffusion equation of suspended sediment is as follows:

$$\frac{\partial hC_z}{\partial t} + \frac{\partial}{\partial x}\left( huC_s - \varepsilon h \frac{\partial C_s}{\partial x} \right) + \frac{\partial}{\partial y}\left( hvC_s - \varepsilon h \frac{\partial C_s}{\partial y} \right) = F_s \tag{15}$$

For the study area which is dominated by cohesive fine-grained sediment, full silting model was used to calculate the sediment erosion and deposition rate which were simulated by the Partheniades–Krone equation [18]:

$$\begin{aligned} F'_s &= E^l - D^l \\ E^l &= M^l S\left(\tau_{\alpha v}, \tau^l_{cr,e}\right) \\ D^l &= w^l_s c^l_b S\left(\tau_{cw}, \tau^l_{cr,d}\right) \\ c^l_b &= c^l \left\{ z = \frac{\Delta z_b}{2}, t \right\} \end{aligned} \tag{16}$$

where $E^l$ represents the scour sediment flux; $M^l$ represents scour coefficient; $D^l$ represents deposition flux; $w^l_s$ represents sediment settling velocity; $c^l_b$ is average sediment concentration near bottom. $S\left(\tau_{\alpha v}, \tau^l_{cr,e}\right)$ and $S\left(\tau_{cw}, \tau^l_{cr,d}\right)$ are respectively expressed in the following equation:

$$S\left(\tau_{cw}, \tau^l_{cr,e}\right) = \begin{cases} \frac{\tau_{cw}}{\tau^l_{cr,e}} - 1, & \text{when } \mid \tau_{cw} > \tau^l_{cr,e} \\ 0, & \text{when } \tau_{cv} < \tau^l_{cr,e} \end{cases}$$

$$S\left(\tau_{cw}, \tau^l_{cr,d}\right) = \begin{cases} 1 - \frac{\tau_{cw}}{\tau^l_{cr,d}}, & \text{when } \tau_{cw} > \tau^l_{cr,d} \\ 0, & \text{when } \tau_{cv} < \tau^l_{cr,d} \end{cases} \tag{17}$$

where $\tau^l_{cr,e}$ represents the critical shear stress of sediment erosion, $\tau^l_{cr,d}$ denotes the critical shear stress of deposition, $\tau_{cv}$ is the shear stress on the bed surface.

The bed sediment transportation will lead to the change of bed elevation which is directly determined by the sediment deposition flux and suspended flux, expressed as:

$$\left(1 - \varepsilon_{por}\right)\frac{\partial z_b}{\partial t} + \frac{\partial S_x}{\partial x} + \frac{\partial S_y}{\partial x} = T_d \tag{18}$$

where $S_x$, $S_y$ are the sediment transport components, zb is the bed elevation, $\varepsilon_{por}$ is the bed porosity, which is generally 0.4. $T_d$ is the bed silting deposition or erosion rate, $T_d = D - E$.

## 3. Results and Discussion

In local scour modeling, the calculation time was set from 8:00 September 10 to 23:00 16 September 2007, which fully included the whole time period of Typhoon Lily. Figure 10a shows the change of flow velocity over time around KP300. The velocity response is not very strong, which is related to a variety of factors, such as the intensity of the typhoon, the type of path, the relationship between its acting time and ebb and flow, and topographic factors [10]. However, the change of the wave is very obvious. During the period of typhoon transit, the effective wave height of the wave near KP300 increases obviously, and decreases with the distance of the typhoon. The extreme effective wave height reached near 3 m at around 18:00 on 15 September (Figure 10b).

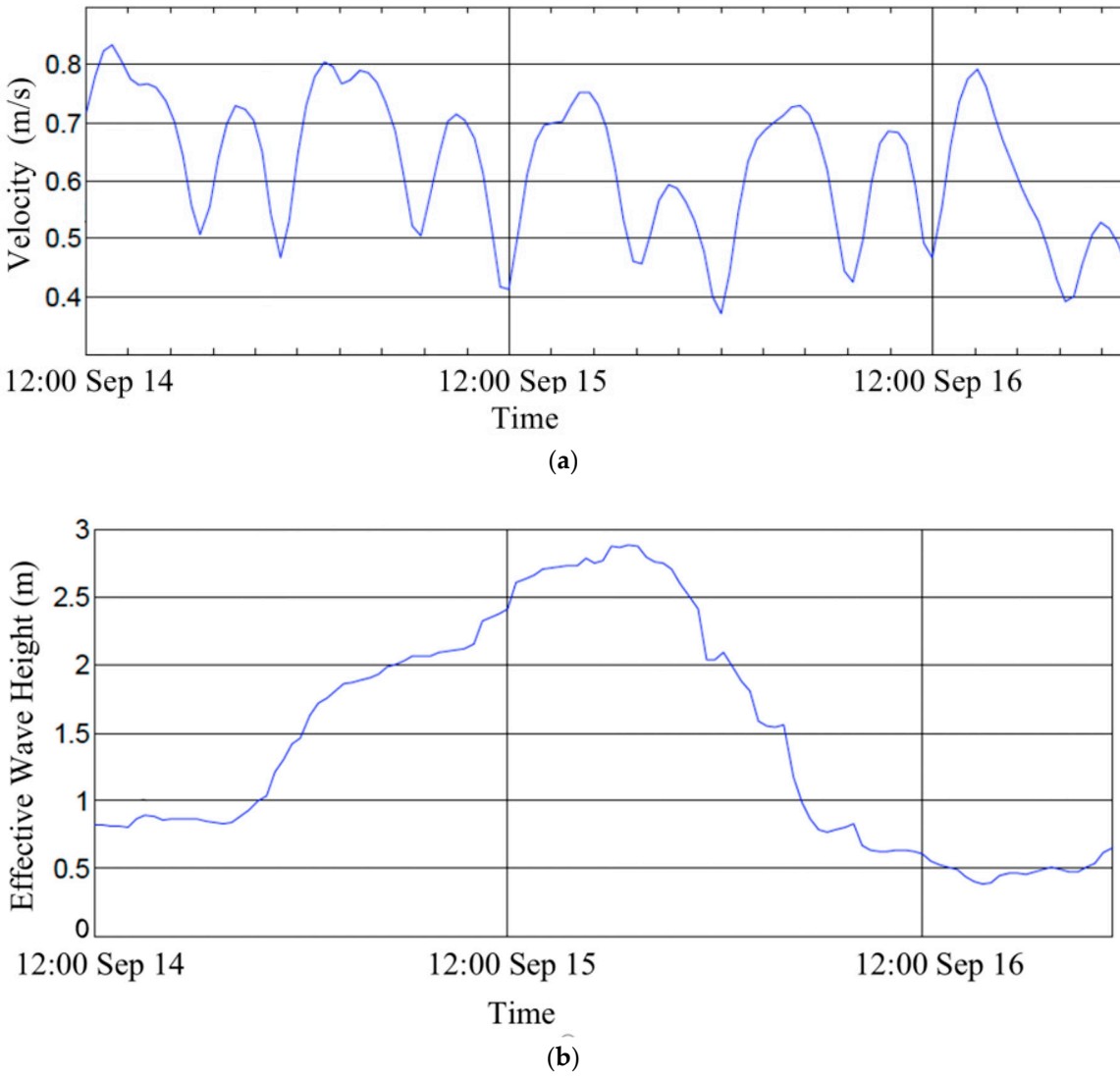

**Figure 10.** The change in flow velocity and effective wave height over time around KP300. (**a**) The flow velocity over time around KP300; (**b**) The effective wave height over time around KP300.

For the study area which is dominated by cohesive fine-grained sediment, the full silting model was used to calculate the sediment erosion and deposition rate which were simulated by the Partheniades–Krone equation [21]:

The results of the local scour modeling are shown in Figures 11 and 12. As can be seen from Figure 11, four points (a, b, c, d) on the near-shore section of the pipeline route are selected to obtain their scour and deposition ranges during the typhoon. It can be intuitively seen from Figure 11 that an area centered on the location of the KP300 pipeline (shown by the red circle in Figure 11) experienced an obvious scouring process during the typhoon. However, in the area east of the pipeline route KP280, with the continuous increment of water depth, the impact of waves on the seabed bottom becomes smaller, and the topography becomes uncomplicated, so the flow structure becomes simple and the scour effect is not obvious.

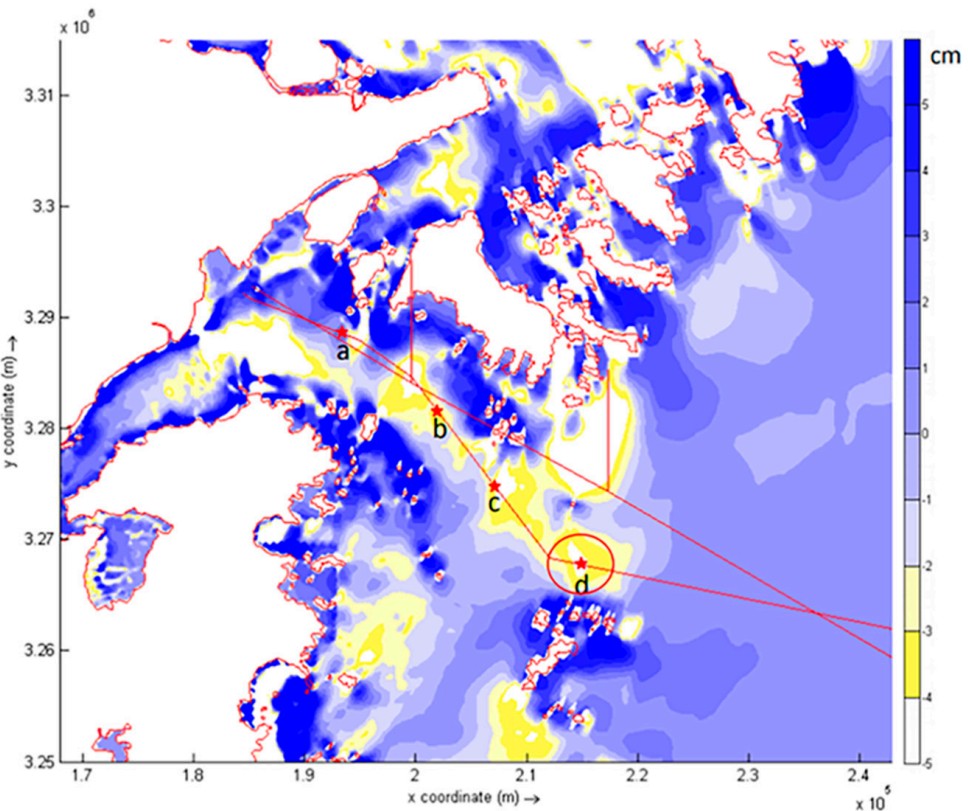

**Figure 11.** Regional Erosion and Deposition after Typhoon Lily.

Figure 12 shows the scouring process at pipelines for four different points. The scour and deposition of points a, b, c and d can be summarized as silting, balance, scour and intense scour, showing a transitional trend. Explaining it in detail, point a is silted up during the typhoon, point b is basically in equilibrium, point c is scoured during the most violent period of the typhoon and then slowly silted up, and point d (near KP300) has a relatively violent scour process during the typhoon. After the typhoon, it begins to slowly accumulate so as to recover to the original state and reach equilibrium.

A strong current will be formed when the typhoon comes. As a result, the shear stress at the bed bottom will increase rapidly, resulting in a large number of sediment movement. Typhoons transfer energy to a water body and make it possible to scour violently in a short time by changing the shear stress of the bed and the sediment-carrying capacity of water. From point a to point d, these four observation areas are far away from the shore (Figure 11). The sediment cannot be supplied in time for areas such as point c and point d, which are relatively far away from the shore, resulting in severe regional scouring, exposure, and even suspension of pipelines. Restricted by the terrain, the flows and wind waves are roughly in the same direction. In areas (point c and point d) that are scoured during a typhoon, sediment from these areas is transported by the current and deposited in an adjacent area (point a) along the path of the typhoon. Meanwhile, in the middle area, a balance between

scouring and deposition appears (point b). As a result, the original dynamic balance of scouring and silting was ruined which led to severe local scouring within a short period of time, resulting in critical suspension of pipelines and other dangerous conditions.

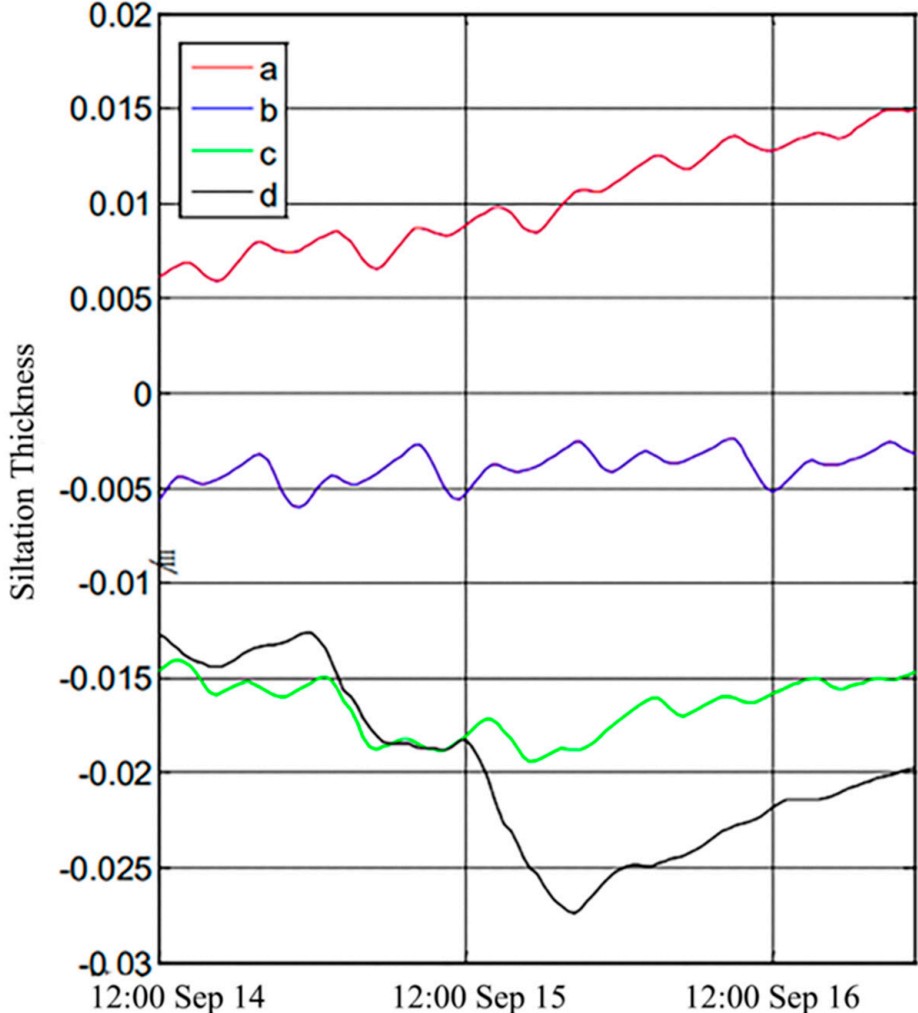

**Figure 12.** Scouring process at pipelines for different points.

Figure 13 shows the shallow stratum profile of the suspended pipeline detected in 2007. The reflection wave of the suspended pipeline and the V-shaped trench scoured by the bottom current are clearly visible, and the V-shaped trench is asymmetric on both sides of the pipeline. The south slope is gentle, while the north slope is steep, which has to do with the direction of the bottom flow. From the scour pit shape, the scour is mainly caused by the south-to-north bottom flow, and a long gentle slope on the left side of the pipeline is formed under the influence of the wake flow. Seabed sediment near KP300 is unconsolidated clay soil, so the critical starting velocity is low, and the water depth is shallow. Therefore, the wind wave has a great influence on the bottom shear stress, providing conditions for the scouring of pipelines near KP300. Combined with mathematical model results and theoretical analysis, it is reasonable to believe that the main culprit behind submarine pipeline suspension danger in the Chunxiao gas field group was the typhoon weather with high density and intensity in 2007.

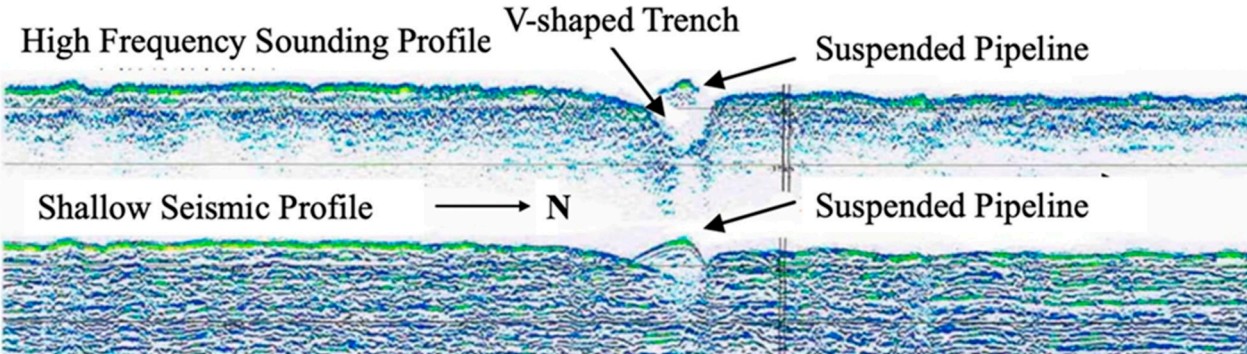

**Figure 13.** Shallow profile of suspended pipelines.

Under the influence of Typhoon Lily in 2007, the original dynamic equilibrium of scouring and deposition was ruined in the offshore section of submarine pipeline in the Chunxiao Gas Field Group, which led to severe local scouring in a short time, resulting in critical suspension and a suspension of the pipeline in this section. Energy was transferred from the typhoons to the water body by changing the bed shear stress and sediment-carrying capacity of water body, making it possible to scour intensely in a short time.

According to the monitoring data for many years, it is concluded that the main reason for the local suspension of the pipeline in 2007 is the high-intensity typhoon weather in 2007. Based on the wave current coupled numerical model, the erosion and deposition of the pipeline route area after Typhoon Lily in 2007 were calculated. The calculation results show that the area near KP300 where the suspended pipeline was found in 2007 is the most serious scour, which also verifies the above inference.

## 4. Conclusions

A numerical model of storm waves in the coastal area of Zhejiang is established. The calculated wind and wave fields caused by typhoon 0712 are in good agreement with the calculated results of NOAA WAVEWATCH III, which indicates that the model can reflect the distribution of wind and waves in this region under typhoon conditions. Analysis and calculation show that under extreme weather conditions, the contribution of strong waves to the bottom shear stress cannot be ignored. Only when the influence of waves is considered, the actual situation can be reflected in the calculation of local scour.

The contribution of waves (especially under extreme conditions) to the bottom shear stress is not to be ignored. In this paper, a numerical model of storm waves in the coastal area of Zhejiang is established. The calculated wind and wave fields caused by typhoon 0712 are in good agreement with the calculated results of NOAA WAVEWATCH III, which indicates that the model can reflect the distribution of wind and waves in this region under typhoon conditions. Because the area near the submarine pipeline KP300 of the Chunxiao gas field group is located at the entrance of tidal channel, when the typhoon storm comes, it will inevitably cause a strong current. Restricted by the terrain, the flow, wind and waves are roughly in the same direction. At this time, the shear stress at the bottom of the bed will increase sharply, and a large amount of sediment will be lifted. Because the area is relatively far from the shore, the sediment cannot be replenished in time, and the sediment deposition is far from offset by the amount of scouring under extreme conditions, which leads to regional scouring at a rapid rate, resulting in exposed or even suspended pipelines, endangering their safety.

Coupled with the hydrodynamic model and wave model under the action of the typhoon, and the calculations around typhoon 0712, the results obtained from the coupling model are provided to the sediment model to obtain the regional scour distribution under the typhoon. The results show that the area around KP300 shows strong scour under typhoon action, and the calculation results are consistent with the monitoring results of the pipeline. It shows that the typhoon has an impact on the safety of submarine pipelines

in this area. It is the first time in China that a numerical model is used to study the safety condition of submarine pipelines in a specific area, specific environment and specific extreme natural conditions, considering the combined action of wave and current. This new method can provide a basis for the selection of submarine pipelines in complex terrain and hydrodynamic environment, and also provides a reference for the monitoring of submarine pipelines.

Based on the monitoring data of submarine pipelines in the Chunxiao gas field group after the typhoon in 2007, it was observed that the suspension section of the pipeline was near KP300. Based on the analysis of topography, marine hydrology, meteorological conditions and submarine sediment, it was inferred that the strong scour and pipeline suspension near KP300 were caused by the typhoon. Then, the pipeline of KP287.66~KP301.906 was selected as the study area, and a numerical model was established to calculate the scouring and deposition condition of the pipelines after Typhoon Lily, a typical typhoon in 2007. Simulation results show that the most serious scouring occurs around KP300, where suspended pipelines were found in 2007 monitoring. According to the monitoring data of the submarine pipelines of the Chunxiao Gas field group for many years combined with the comprehensive environmental factors such as hydrology, sediment, meteorology and topography, it is concluded that the main reason for the partial suspension of the pipeline is the typhoon weather with high intensity and density. On the one hand, this new method can provide a sound basis for line selection of submarine pipelines under complex conditions. On the other hand, it can also provide a reference for submarine pipeline monitoring.

**Author Contributions:** Conceptualization, P.H. and H.D.; methodology, P.H.; investigation, P.H.; resources, H.D.; data curation, P.H.; writing—original draft preparation, H.D.; writing—review and editing, H.D. and L.C.; visualization, X.M.; supervision, H.D.; project administration, P.H.; funding acquisition, H.D. All authors have read and agreed to the published version of the manuscript.

**Funding:** This research was financially supported by the Science and Technology Program Project of ZhouShan City of China (No.2020C41064).

**Data Availability Statement:** Some or all data, models, or code that support the findings of this study are available from the corresponding author upon reasonable request.

**Acknowledgments:** The authors would like to thank staffs in Second Institute of Oceanology for their effort in field observation for this research area.

**Conflicts of Interest:** The authors declare no conflict of interest.

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
