# Peer review of "Numerical Modeling of Submarine Pipeline Scouring under Tropical Storms"

_water, doi:10.3390/w13101425_

Round 1

Reviewer 1 Report

  1. The expressions of the continuity equation (1) and the Navier Stokes equation (2), (3) are quite different from those which are described in standard textbooks of fluid dynamics. These forms are simply explained in the first paragraph of the section 2.3.1, Probably the expressions in this paper might be common among researchers working in the same field as this work. However, a paper published in journals should be written so that it can be understood by a wide range of readers to some extent. A little more detailed explanation is desired. At least, the reference of Arakawa C-type grid system should be cited.
  2. The parts from line 135 to line 232 needs to be completely revised. When using a new symbol in an equation, it must be defined near the location of the equation unless it is generally known. Many important symbols are defined far from the lines of equations. The definition of even some important symbols has been ignored. Since a large number of symbols are used in this paper, it is recommended to add a nomenclature besides explanation in the text.
  3. Line 87: the number of average slope should be accompanied with unit.
  4. Line 188: A reference should be cited for SWAN model.
  5. Line 114: 0.005 – 0.75mm. It is better to express the number by the unit micron instead of mm.
  6. 4: Many vertical lines are drawn in Fig.4. What are those lines?
  7. Line 145: It is described that Q represents the role of source and sink. Evaporation is included as one of such source and sink. In general, evaporation is included in Q, but evaporation is not considered in this work. It is better to exclude the word “evaporation” as the term of what Q represents.
  8. Line 210: The word “and” must be deleted.
  9. Line 217: The reference should be cited for “Hesselmann formula”.

Author Response

Point 1: “1. The expressions of the continuity equation (1) and the Navier Stokes equation (2), (3) are quite different from those which are described in standard textbooks of fluid dynamics. These forms are simply explained in the first paragraph of the section 2.3.1, Probably the expressions in this paper might be common among researchers working in the same field as this work. However, a paper published in journals should be written so that it can be understood by a wide range of readers to some extent. A little more detailed explanation is desired. At least, the reference of Arakawa C-type grid system should be cited.”

Response 1: Thanks for reviewer’s suggestion and question. In this paper, the hydrodynamic model was established by Delft3D, and the equations in this paper used the form in the Delft3D Flow Manual. In order to maintain the uniformity of the paper based on Delft3D, it is recommended to adopt the current equation form[1].

Point 2: “2. The parts from line 135 to line 232 needs to be completely revised. When using a new symbol in an equation, it must be defined near the location of the equation unless it is generally known. Many important symbols are defined far from the lines of equations. The definition of even some important symbols has been ignored. Since a large number of symbols are used in this paper, it is recommended to add a nomenclature besides explanation in the text.”

Response 2: Thanks for reviewer’s suggestion. We are sorry that we made such mistakes. We have revised the parts from line 155 to line 315.

Point 3: “3. Line 87: the number of average slope should be accompanied with unit.”

Response 3: Thanks for your suggestion. It’s our negligence to make this mistake.

It has been revised in manuscript. Please see Line 86.

Line 86:

“The topography of the seabed in the whole area is basically floating without severe undulations or steep ridges, whose average slope is 0.3×10-3.”

Point 4: “4. Line 188: A reference should be cited for SWAN model.”

Response 4: Thanks for your suggestion. We have cited SWAN model in our manuscript. Please see lines 61-76.

Lines 61-76:

“Western European countries began to study the third generation of wave numerical models in 1985, and a WAM group was established to develop a new wave numerical model with a more comprehensive consideration of the source functions in the energy balance equation. Among them, WAM model and Wavewatch â…¢ model are the representatives of the third generation of wave numerical models[2]. Later, Booij et al. summarized and modified the third-generation model, especially the results of WAM parameterization, and proposed SWAN, which is suitable for the computation of offshore waves. After continuous improvement, SWAN has been widely applied in the field of offshore Marine science and engineering. Although the same wave model is the third generation, there are also differences in performance and applicability. In order to foster strengths and circumvent weaknesses, nested applications of different modes have appeared [3]. For example, using Wavewatch â…¢ to build the global wind and wave model and nested SWAN model in the local nearshore area, SWAN can improve the accuracy after obtaining better boundary data. In the simulation of near-shore wind and wave, SWAN often has a better effect than WAVEWATCH â…¢.”

Point 5: “5. Line 114: 0.005 – 0.75mm. It is better to express the number by the unit micron instead of mm.”

Response 5: Thanks for reviewer’s suggestion. We have revised our manuscript in Lines112-115.The revised part is as follows:

Lines 112-115:

“The sediment was mainly composed of silt and clay, of which clay was slightly dominant, accompanied by a small amount of shell fragments, and the sediment particle size was between 5 ~ 75m.”

Point 6: “6. 4: Many vertical lines are drawn in Fig.4. What are those lines?”

Response 6: The vertical line is the drilling position.

Point 7: “7. Line 145: It is described that Q represents the role of source and sink. Evaporation is included as one of such source and sink. In general, evaporation is included in Q, but evaporation is not considered in this work. It is better to exclude the word “evaporation” as the term of what Q represents.”

Response 7: Thanks for your suggestion. It’s our negligence to make this mistake.

It has been revised in manuscript. Please see Line 147.

Line 147:

“Q represents the role of source and sink, such as drainage and precipitation.”

Point 8: “  8. Line 210: The word “and” must be deleted.”

Response 8: Thanks for reviewer’s suggestion. We have revised our manuscript. Please see Line 209.

Line 209:

“The input of wind energy is described by resonance mechanism feedback mechanism in Swan model:”

Point 9: 9. Line 217: The reference should be cited for “Hesselmann formula”.

Response 9: Thanks for reviewer’s suggestion. We have revised our manuscript.

Please see line 281 in revised manuscript.

Line 281:

“In SWAN, wave spectrum is described by the spectral action balance equation given in Hesselmann[20].”

Thanks again!

We tried our best to improve the manuscript and made some changes in the manuscript.

These changes will not influence the content and framework of the paper. We carefully edited English usage, grammar, punctuation, spelling, figures and overall styles in the revised version. We appreciate for Reviewers’ warm work earnestly, and hope that the correction will meet with approval. Once again, thank you very much for your comments and suggestions.

References: 

  1. ARAKAWA, A.; LAMB, V.R. Computational Design of the Basic Dynamical Processes of the UCLA General Circulation Model; ACADEMIC PRESS, INC., 1977; Vol. 17;.
  2. Hasselmann, K.; Hasselmann, S.; Bauer, E.; Janssen, P.A.E.M.; Komen, G.J.; Bertotti, L.; Lionello, P.; Guillaume, A.; Cardone, V.C.; Greenwood, J.A.; et al. The WAM model - a third generation ocean wave prediction model. J. Phys. Ocean.1988, 18, 1775–1810.
  3. Booji, N.; Holthuijsen, L.H.; Ris, R.C. The SWAN wave model for shallow water. Proc. 25th Int. Conf. Coast. Eng.1996, 668–676.

Reviewer 2 Report

Specific comments are provided in a PDF file.

General comments

  • Authors need to clearly define the limitations of the study and then support their points with the relevant literature. For example -what are the reasons behind the variation of results of this work with WAVEWATCH data (Fig. 8) - views or observations of Authors should be supported by the literature
  • The conclusion section of the manuscript at present is not precise enough. It needs to be rewritten with the presentation of key findings and impacts of this research.

Author Response

Point 1: “Authors need to clearly define the limitations of the study and then support their points with the relevant literature. For example -what are the reasons behind the variation of results of this work with WAVEWATCH data (Fig. 8) - views or observations of Authors should be supported by the literature”

Response 1: Thanks for reviewer’s question and suggestion. Although the two do not correspond well in terms of magnitude, they are very consistent in terms of the trend of effective wave heights, and the peaks of the three points all appear at the same time. It shows that the model can reflect the wave response in the coastal area of Zhejiang when typhoon passes by. For this paper, the ultimate purpose is not to study the platform wind and wave itself, but to consider the influence of waves on the seabed shear stress and scour effect. Therefore, such a result is acceptable. In addition, the difference of effective wave heights between the two is also caused by many reasons, some of which are even unavoidable. First of all, although both belong to the third generation numerical wave model, there are theoretical differences, especially in the nearshore, the processing of various physical processes is not the same, and the input of wind energy is not consistent. Secondly, none of the three boundaries of SWAN model in this paper can provide real boundary conditions, which will certainly affect the calculation accuracy. Finally, NOAA's Wavewatch â…¢ forecast value is for the whole world, with a low resolution. Due to the topography and other factors, the accuracy of the near shore is relatively poor. For the regions affected by the typhoon field, there is no detailed wind field, which is bound to affect the calculation accuracy.

Point 2: “The conclusion section of the manuscript at present is not precise enough. It needs to be rewritten with the presentation of key findings and impacts of this research.”

Response 2: Thanks for reviewer’s suggestion. A large-scale numerical model considering wave current coupling is established for the coastal area of Zhejiang Province. Both analysis and calculation show that under extreme weather conditions, the contribution of strong waves to the bottom shear stress cannot be ignored. Only when the influence of wave is considered, the actual situation can be reflected in the calculation of local scour.

Coastal areas are densely populated and economically developed, and development and utilization activities are increasingly frequent. The construction of coastal aquaculture, coastal tourism and economic development zones is booming. The ocean has brought the coastal area the superior environment and the good development condition, the coastal area has already formed the coastal economy zone which relies on the ocean initially. Therefore, under the background of the booming Marine economic construction, it is of great practical significance to improve the disaster prevention and mitigation of Marine engineering, especially to strengthen the research on Marine disasters under extreme weather conditions with strong destructive power. Although the extreme hydrodynamic environment caused by typhoons is only a part of the damage to the submarine oil and gas pipelines, it should be paid more attention to in view of the complexity of the construction and maintenance of the submarine pipelines and the important role they play in ensuring the national energy supply.

Because the area near the submarine pipeline KP300 of Chunxiao gas field group is located at the entrance of tidal channel, when the typhoon storm comes, it will inevitably cause strong current. Restricted by the terrain, the flow and the wind and waves are roughly in the same direction. At this time, the shear stress at the bottom of the bed will increase sharply, and a large amount of sediment will be lifted. And because the area is relatively far from shore, the sediment cannot be replenished in time, and the sediment deposition is far from offset by the amount of scouring under extreme conditions, which leads to regional scouring at a rapid rate, resulting in the exposed or even suspended pipelines, endangering their safety. In 2007, Typhoon 0712 Neri was followed by Typhoon Wipha 0713, a high density of typhoon that was clearly more harmful than a single typhoon. In coastal and shallow water areas, a high-energy event in the water body, such as a typhoon, hurricane, storm surge, or earthquake and tsunami, can wash away deposits that have been formed over a long period of normal weather conditions. As discussed in this paper, the original dynamic balance of scour and silt was broken in the offshore section of the offshore pipeline of Chunxiao Gas Field Group under the influence of Typhoon Nili, Weipa and Rosa in 2007, resulting in a short period of intense local scour, resulting in critical suspension and other dangerous conditions in this section of pipeline. The typhoon transfers energy to the water body, which makes the severe scouring in a short time possible by changing the shear stress of the bottom bed and the sediment carrying capacity of the water body. In shallow sea area, submarine topography and coastal shape are important boundary conditions of ocean dynamic processes such as tidal current, and all elements interact and influence each other. Under unconventional conditions, the balance is broken and a certain element is forced to change. When the external conditions return to normal, other elements will force this element to return to its original state so as to achieve a new balance. Here, areas that were washed out during the storm will develop towards siltation later on. This is why in the monitoring of 2009, the nearshore section of the pipeline area has silted up compared with 2007, and the pipeline is in a safe state.

This paper draws the following conclusions:

(1) A three-dimensional high-resolution numerical model of storm surge along the coast of Zhejiang Province was established. The calculation accuracy is very high; Storm water increase and storm flow under the influence of Typhoon Rammasun 0205 are calculated, which are in good agreement with the measured values. The model has the ability to forecast storm surge for the coastal area of Zhejiang Province in China.

(2) The contribution of waves (especially under extreme conditions) to the bottom shear stress is not slighted. In this paper, a numerical model of storm waves in the coastal area of Zhejiang is established. The calculated wind and wave fields caused by typhoon 0712 are in good agreement with the calculated results of NOAA Wavewatch â…¢, which indicates that the model can reflect the distribution of wind and wave in this region under typhoon conditions.

(3) Coupled the hydrodynamic model and wave model under the action of typhoon, and calculated the typhoon 0712.The results obtained from the coupling model are provided to the sediment model to obtain the regional scour distribution under typhoon. The results show that the area around KP300 shows strong scour under typhoon action, and the calculation results are consistent with the monitoring results of the pipeline. It shows that typhoon has an impact on the safety of submarine pipelines in this area. It is the first time in China that the numerical model is used to study the safety condition of submarine pipeline in specific area, specific environment and specific extreme natural conditions, considering the combined action of wave and current. This new method can provide a basis for the selection of submarine pipelines in complex terrain and hydrodynamic environment, and also provide a reference for the monitoring of submarine pipelines.

In addition, we revised some parts according to your suggestion on PDF. Here are  reviewer’s suggestions and revised parts.

  1. The figure 4 has been revised. Please see Figure 4 in the revised manuscript.

Figure 4. Zonation Map of Submarine Pipeline Routing Area

  1. The reviewer said that:“Section 2.2.3 - Authors should clearly define all the parameters, such as MHW, MLW and etc. Also Wave Rose is needed to be explained here.  2.2.4 - It is better to provide a Wind Rose for the particular area

Response: We are sorry to say that we don’t have that statistics.

  1. This figure needs to be improved. Authors are requested to improve the visualization of the path of the Typhoon.

Response: Thanks for reviewer’s suggestion. We have improved the visualization of the path of Typhoon.

  1. That's needs to be further explained with the support of relevant literature.

Response: Thanks for reviewer’s suggestion. We have added some relevant literature.

Please see Introduction in Line 21-86.

Lines 21-86:

“Oil and gas pipelines are regarded as the lifelines of marine energy transmission. Since the first submarine pipeline was laid in the Mexico Gulf by the United States in 1954, the bottom of the sea has seen the vigorous development of submarine pipelines all over the word[1]. Nowadays, more than 3000 kilometer of submarine pipelines have been laid in China. Along the coast of Zhejiang Province, submarine pipelines of groups of Chunxiao Gas Field, Pinghu Oil and Gas Field and Hangzhou Bay Cross-sea Oil have been constructed[2].

The pipeline of Pinghu Oil and Gas Field landed in Zhoushan City, Zhejiang Province where two pipelines broke on 2000 (Chen et al. 2005) [3], leading to the oil spills. As a result, hundreds of millions of dollars were spent, and serious environmental damage was caused by widespread oil pollution. Therefore, it is essential to ensure the safety of submarine pipelines. Figure 1 shows how accidents happened due to the local scour at pipelines.

In 1982, six of European leading gas storage and transport companies launched a campaign to collect data on accidents in pipeline systems, setting up the European Pipeline Accident Data Organization (EGIG), which made statistics on the causes of the failure of the submarine gas pipelines between 1970 and 2010 [4]. Now EGIG is a co-operation between a group of seventeen major gas transmission system operators in Europe and is the owner of an extensive gas pipeline-incident database. The creation of this extensive pipeline-incident database (1982) [5]has helped pipeline operators to demonstrate the safety performances of Europe's gas pipelines, which is useful for the pipeline operators to improve safety in their gas pipeline transmission systems [6](EGIG,2018).

Scouring at pipelines is affected by many environmental factors such as topography, submarine sediments, marine hydrology and wind fields [7][8][9][10][11]. The strong hydrodynamic disturbance, caused by typhoon destroys the equilibrium state of seabed sediments under the action of tidal wave, which usually takes long time to form. At the peak of the strongest storm action, the currents strongly scour the seabed, carrying a large amount of sediment in water, which is transported to the sea with the ebb tide of the storm surge, thus affecting the laying state of the undersea pipelines [2][5][12][13].

Application of numerical simulation in scouring and silting, such as the shear stress calculation model of horizontal seabed bed surface, model of sediment transport under storm conditions, etc., has increased tremendously in recent years for analyzing different environmental factors[14][15][16][17]. These mathematical models simulate the scouring and silting situation near the pipeline under certain natural factors, which can reveal the local scouring and silting law. However, there are few studies on the sediment module based on the wave-current coupling model, and the changes of waves, tidal currents and sediment caused by typhoons are not fully revealed.

Western European countries began to study the third generation of wave numerical models in 1985, and a WAM group was established to develop a new wave numerical model with a more comprehensive consideration of the source functions in the energy balance equation. Among them, WAM model and Wavewatch â…¢ model are the representatives of the third generation of wave numerical models[18]. Later, Booij summarized and modified the third-generation model, especially the results of WAM parameterization, and proposed SWAN, which is suitable for the computation of offshore waves. After continuous improvement, SWAN has been widely applied in the field of offshore Marine science and engineering. Although the same wave model is the third generation, there are also differences in performance and applicability. In order to foster strengths and circumvent weaknesses, nested applications of different modes have appeared [19]. For example, using Wavewatch â…¢ to build the global wind and wave model and nested SWAN model in the local nearshore area, SWAN can improve the accuracy after obtaining better boundary data. In the simulation of near-shore wind and wave, SWAN often has a better effect than WAVEWATCH â…¢.

In this research, a large area numerical model was established with the consideration of the interaction between wave and water flow, which simulated the dynamic elements of the ocean under typhoon. Meanwhile, the scouring and silting conditions of the offshore area with complicated topography during typhoon was considered, and the rules of scouring and silting in the area where the submarine pipeline is studied. Considering currents and wave, the numerical model is used to study the safety of submarine pipelines in specific areas and specific extreme natural conditions. This new method can provide a sound basis for line selection of submarine pipelines under extreme conditions, such as complex terrain, geomorphologic region and hydrodynamic environment, and can also provide reference for the monitoring of submarine pipelines.”

  1. Have you defined Effective Wave height?

Response: Thanks for reviewer’s question. It’s our negligence to make this mistake.

We have revised it in manuscript. Please see lines 575-585.

“Effective wave height refers to the actual wave height calculated according to certain rules. Since sea waves are actually a random combination of waves with different wave heights, periods and directions, the wave height value of a wave is not representative. For this reason, in any wave group composed of n waves, the wave heights in the wave train are arranged in order from large to small, and the first 1/3 waves are determined as effective waves. The wave height and period of the effective wave are equal to the average wave height and average period of N over 3 waves[19].”

  1. Authors are requested to rewrite the Conclusions, particularly the highlighted sentences. It is not clear I suppose at present, what this study brings to the scientific community?

Response: Thanks for reviewer’s suggestion. We have rewritten the Conclusions, particularly the highlighted sentences. Please see Lines 542-562.

Conclusions

A numerical model of storm waves in the coastal area of Zhejiang is established. The calculated wind and wave fields caused by typhoon 0712 are in good agreement with the calculated results of NOAA Wavewatch â…¢, which indicates that the model can reflect the distribution of wind and wave in this region under typhoon conditions.

Based on the monitoring data of submarine pipelines in Chunxiao gas field group after typhoon in 2007, it was observed that the suspension section of the pipeline was near KP300. Based on the analysis of topography, marine hydrology, meteorological conditions and submarine sediment, it was inferred that the strong scour and pipeline suspension near KP300 were caused by typhoon. Then, the pipeline of KP287.66 ~ KP301.906 was selected as the study area, and a numerical model was established to calculate the scouring and deposition condition of the pipelines after Typhoon Lily, a typical typhoon in 2007. Simulation results show that the most serious scouring occurs around KP300, where suspended pipelines were found in 2007 monitoring. According to the monitoring data of the submarine pipelines of Chunxiao Gas field group for many years combined with the comprehensive environmental factors such as hydrology, sediment, meteorology, topography,it is concluded that the main reason for the partial suspension of the pipeline is the typhoon weather with high intensity and density. On the one hand, this new method can provide a sound basis for line selection of submarine pipelines under complex conditions. On the other hand, it can also provide a reference for submarine pipeline monitoring.

Thanks again!

We tried our best to improve the manuscript and made some changes in the manuscript.

These changes will not influence the content and framework of the paper. We carefully edited English usage, grammar, punctuation, spelling, figures and overall styles in the revised version. We appreciate for Reviewers’ warm work earnestly, and hope that the correction will meet with approval. Once again, thank you very much for your comments and suggestions.

Reviewer 3 Report

The authors studied the sediment transport under the action of waves and currents in east China, established with a numerical model of sediment scouring and deposition 12 combining wave and currents. They found that typhoon weather with high intensity and density will lead to suspension of pipelineswith local scour. The findings from the manuscript does attract special attentions of the readers of the journal; hence the paper should be published after minor revision. My revision requests are listed as follows.

  1. Describe the modeling condition in detail.
  2. Describe the weather condition for modeling to be shown clearly.
  3. In conclusion, summarize the main reason of such a phenomena of sediment transport in east China. 

Author Response

Point 1: “The authors studied the sediment transport under the action of waves and currents in east China, established with a numerical model of sediment scouring and deposition 12 combining wave and currents. They found that typhoon weather with high intensity and density will lead to suspension of pipeline swith local scour. The findings from the manuscript does attract special attentions of the readers of the journal; hence the paper should be published after minor revision. My revision requests are listed as follows.

Response 1: We are appreciated the positive comments given by the reviewer. We have studied comments carefully and made a thorough proofreading which we hope meet with approval.

Point 2: 1. Describe the modeling condition in detail.

Response 2: Thanks for reviewer’s suggestion. The calculation time for Typhoon Rammasun 0205 is from 8 o 'clock on June 30, 2002 to 0 o 'clock on July 11, 2002 Beijing time, with a time step of 2min. In order to save time, the vertical average is divided into 6 layers, the gravitational acceleration is set as 9.81m/s², and the density of sea water is 1025kg/m². The air density is 1.293kg/m², the horizontal vortex viscosity is 25 m²/s, and the seabed roughness coefficient, which is sensitive to the calculation results, is characterized by Manning coefficient according to the regional characteristics, and the empirical formula which has good application effect in this region is selected to calculate. That is, m=0.015+0.00255× E15-4) (formula adopted by Nanjing Hydraulic Research Institute), where m is the Manning coefficient and depth is the actual water depth at the grid.

Here is initial conditions and boundary conditions

Initial state: 0m relative to the mean sea level, no wind field on the sea surface, pressure of 1 atmosphere. Boundary conditions: land and water boundary, rigid wall treatment, that is, try to flow to zero. Boundaries of outer sea water, from the whole

The spherical tidal wave model TPX06, which is driven by eight sub-tides of M, S, N, K, K, O, P and Q, can basically construct the real astronomical tidal process in the deep sea. The calculation area of river boundary mainly includes two large rivers, the Yangtze River and the Qiantang River. The given constant flow in the upper reaches is taken as the boundary. According to the historical statistical data of many years, the flow of the Yangtze River estuary is set as 40,000 m2/s, and that of the Qiantang River estuary is set as 4,000 m2/s. The flood plain process of storm surge and flow is processed through the judgment of dry and wet grid.

Point 3: 2. Describe the weather condition for modeling to be shown clearly.

Response 3: Tropical cyclone is the main catastrophic weather system affecting the coastal waters of Zhejiang Province. When it hits, it is often accompanied by strong wind, heavy rain, large waves and storm surge, etc., which will cause serious threats to the coastal engineering construction projects. Statistics for recent decades indicate that the average number of tropical cyclones affecting the region is about four per year. The most affected years were 1959 and 1960, with 8. The least affected year was 1996, which was almost not affected by tropical cyclones. Tropical cyclones generally occur from May to November, and are mainly concentrated in July to September, accounting for about 80% of the total, of which August is the peak of tropical cyclone activity, accounting for 35% of the total. In recent decades, the coastal areas of Zhejiang Province were affected by tropical cyclones on May 19 at the earliest (Typhoon 6103), and the end was on November 18 at the latest (Typhoon 6721). There are three main types of tropical cyclones in terms of their moving paths. Firstly, it landed in the central and southern part of Zhejiang, and then turned to the northeast to die out (D11) or turned to the northwest inland to die out (D12), which are collectively referred to as D1 class. Second, it landed between the Zhejiang-Fujian border and Xiamen, and then turned to the northeast to go to sea and died out (D21) or turned to the northwest inland to die out (D22), which are collectively referred to as D2. The third is the sea skim type has DN and HWN class. Typhoon 5612 and 9711, for example, were the D11 types that landed in the central and southern parts of Zhejiang Province. In addition, typhoon D, a type of passing over the sea near Zhoushan Islands, caused serious damage to the coast of Zhejiang, such as typhoon 7413, 8114 and so on.

Point 4: 3. In conclusion, summarize the main reason of such a phenomena of sediment transport in east China. .

Response 4: Thanks for reviewer’s suggestion. Waves lift sand and currents carry it. The interaction among wave, current and bottom is a very common phenomenon in offshore environment. Severe scouring caused by strong waves and strong currents caused by storms can be attributed to two reasons: 1) the shear stress at the bottom of the bed increases, which causes a large amount of seabed sediment to be lifted and mixed in the water; 2) Turbulent water strengthens its own sediment carrying capacity, and strong current and strong sediment carrying capacity make it possible to transport a large amount of sediment outward in a short period of time.

By establishing a coupling large regional model considering the influence of wave convection and the influence of flow on wave, the dynamic elements of the ocean under the action of typhoon can be reduced as much as possible, and the scouring and silting conditions in the offshore area with complex terrain during typhoon period can be investigated, so as to understand the scouring and silting rules in the area where submarine pipelines are located.

This paper emphasizes the influence of waves many times, because waves contribute a lot to the bottom shear stress, and the bottom shear stress is the most critical parameter for the bottom bed scouring process, which is very important for the study of submarine pipeline scouring under typhoon conditions in this paper. In estuarine and coastal areas, the wave-current interaction directly affects the initiation, mixing and transport of sediment. It is a common phenomenon that "waves lift sand and currents transport sediment" in estuaries and coastal areas. The wave action will increase the near bottom velocity gradient and enhance the turbulence, so that the bed surface stress significantly increases compared with that in the case of single flow. Fredsoe & Deigaard have analyzed in detail the changes of boundary layer structure, velocity distribution and eddy viscosity coefficient under the action of wave flow [75]. The strength of the wave-current interaction has a great relationship with the period of the incident wave. Voulgaris & Wallbridge studied the influence of wave period on the interaction of wave and current, and pointed out that when the wave period is small, the wave-current interaction decreases and can be treated as a linear process. With the increase of wave period, the wave-current interaction also gradually increases. The wave-flow interaction presents nonlinear characteristics[1].

Wave action will lead to many changes in the flow, Jing & Peter pointed out that the wave-current interaction will

It results in two significant effects :(1) it causes the change of eddy viscosity coefficient, which leads to the change of flow structure; (2) Increased flow turbulence, resulting in an increase in the bottom shear stress [2]. Among them, the most important effect is the change of bed surface stress, which will directly lead to the initiation of sediment. In order to take this effect into account, many bed surface stress models under the action of wave flow have been proposed, which can be summarized into two categories: the first is the model for iterative solution of wave flow, such as Christoffersen and Jonsson[3]. Grant and Madsen[4], the modified GM model, etc. These models all calculate the bottom friction coefficient under the combined action of wave and flow through iteration. Since such models start from the distribution of viscosity coefficient and velocity structure, and the physical mechanism is relatively clear, they are widely used, such as Hong Zhang [5]. Secondly, the model obtained by directly adding the wave-current components and considering the nonlinearity of the wave-current interaction does not require iterative calculation and is simple in form, which is widely used in the rough calculation of the wave-current interaction.

Thanks again!

We tried our best to improve the manuscript and made some changes in the manuscript.

These changes will not influence the content and framework of the paper. We carefully edited English usage, grammar, punctuation, spelling, figures and overall styles in the revised version. We appreciate for Reviewers’ warm work earnestly, and hope that the correction will meet with approval. Once again, thank you very much for your comments and suggestions.

References: 

  1. Voulgaris, G.; Wallbridge, S.; Tomlinson, B.N.; Collins, M.B. Laboratory investigations into wave period effects on sand bed erodibility, under the combined action of waves and currents. Coast. Eng.1995, 26, 117–134, doi:10.1016/0378-3839(95)00026-7.
  2. Jing, L.; Ridd, P. V. Wave-current bottom shear stresses and sediment resuspension in Cleveland Bay, Australia. Coast. Eng.1996, 29, 169–186, doi:10.1016/S0378-3839(96)00023-3.
  3. Christoffersen, J.B. Bed friction and dissipation in a combined current and wave. 1985, 12.
  4. Teague, W.J.; Jarosz, E.; Keen, T.R.; Wang, D.W.; Hulbert, M.S. Bottom scour observed under Hurricane Ivan. Geophys. Res. Lett.2006, 33, 1–3, doi:10.1029/2005GL025281.
  5. Zhang, H.; Madsen, O.S.; Sannasiraj, S.A.; Soon Chan, E. Hydrodynamic model with wave-current interaction in coastal regions. Estuar. Coast. Shelf Sci.2004, 61, 317–324, doi:10.1016/j.ecss.2004.06.002.

Round 2

Reviewer 1 Report

The authors have answered all the questions of the reviewer and have written their comments in authors’ reply, but some of them are not reflected in the manuscript of the revised version. Corrections are not found in the revised version though they have shown corrections in their reply. Such examples are as follows:

Point 3, Point 7, Point 8 and Point 9.

Author Response

Dear Reviewer:

Thank you for your valuable suggestions! The revison has been made and this part is highlighted in the manuscript. We are looking forward to your precious advice.

 Yours,

Haiyang Dong

Reviewer 2 Report

While authors have provided somewhat responses to all the comments, they did not revise the manuscript accordingly (such as Points 1 & 2). Once again, I suggest a major revision that should clearly reflect the changes within the manuscript instead of providing the responses only. I would like to see the paper after the recommended major changes. 

Author Response

Dear Reviewer:

Thank you for your valuable suggestions! The revison has been made and this part is highlighted in the manuscript. The conclusion has been revised to provide more details about this research and to express why this kind of phenomenon would happen in East China Sea. And We are looking forward to your precious advice.

 Yours,

Haiyang Dong

Round 3

Reviewer 1 Report

The manuscript has been revised properly.

Reviewer 2 Report

Thank you for the revision.